# Determinants of Physical Activity in Older Adults: Integrating Self-Concordance into the Theory of Planned Behavior

**DOI:** 10.3390/ijerph18115759

**Published:** 2021-05-27

**Authors:** Paula Stehr, Constanze Rossmann, Tabea Kremer, Johanna Geppert

**Affiliations:** Department of Media and Communication Science, University of Erfurt, 99089 Erfurt, Germany; constanze.rossmann@uni-erfurt.de (C.R.); tabea.kremer@uni-erfurt.de (T.K.); johanna.geppert@bfr.bund.de (J.G.)

**Keywords:** physical activity, older adults, theory of planned behavior, self-concordance, self-determination theory, habitual behavior

## Abstract

Based on the theory of planned behavior (TPB), augmented by the concept of self-concordance (derived from self-determination theory, SDT), we conducted a study to identify the key determinants of physical activity in older adults. We applied structural equation modeling of telephone survey data from a random sample of adults aged 65 years and older living in Germany (*N* = 865). Relations of attitude, subjective norms, and perceived behavioral control (PBC) with intention strength and self-concordance of intention to be physically active were tested. Habit strength was analyzed as a moderator. Data analysis showed this model to be well-suited for explaining the intention to be physically active—especially for people with a weak habit. The influence of TPB components on intention would have been underestimated if we had investigated intention strength only, without considering the self-concordance of intention. While attitude and PBC had positive relations with a strong and self-determined intention, the subjective norm showed no relation with intention strength but, rather, with non-self-determined regulation forms. We conclude that the combined model provides a better theoretical foundation from which to explain physical activity intentions than does just one of the theories.

## 1. Introduction

While the proportion of the German population over 67 years of age accounted for 19% of the total population in 2018, it will increase to a proportion of 24 to 30% in 2060 [1]. Against the backdrop of this demographic change, health promotion will become more relevant to the older adult population. Prevalence and incidence rates of most diseases—e.g., cardiovascular diseases, diseases of the musculoskeletal system, diabetes, and malignant growth—are higher among older adults than they are among younger adults. Moreover, multimorbidity increases with age [2].

A moderate increase in physical activity can prevent or at least delay many of those diseases and reduce mortality [3,4,5]. Simultaneously, it can promote mobility and participation in social life. Consequently, the promotion of physical activity is a promising strategy for increasing the number of healthy years for older adults and for preventing diseases and their associated constraints [6].

However, commonly, physical activity levels are too low [7]. Therefore, health promotion strategies are needed that can effectively increase physical activity behavior. People’s physical activity is influenced by individual, social environmental, and physical environmental factors [8]. Accordingly, interventions can be classified into informational (e.g., mass media campaigns), behavioral and social (e.g., social support in community settings), and environmental and policy interventions [9]. This paper focuses on informational and behavioral approaches addressing the individuum. An important prerequisite for effective health campaigning is to identify the determinants driving health behavior [10,11]. Numerous studies refer to the theory of planned behavior (TPB; [12]) in order to explain (health) behavior and its determinants as a background for theory-based campaigning. Several meta-analyses confirm most assumptions of TPB in the health domain as well as in other contexts [13,14,15,16] (for an overview, see [12,17]). However, in the context of physical activity among older adults, research often finds significant effects of attitude and perceived behavioral control (PBC) on intention, but no effect of the subjective norm on intention [18,19]. This indicates that it might be fruitful to complement the theory to improve the predictability of this specific behavior. While subjective norms are only weakly related to behavioral intention, studies show that they are related to another antecedent of behavior—the self-determination of motivation [20]. In a similar direction, psychologists argued that behavioral intention is composed of two aspects: the strength of the intention and its self-concordance [21]. “Whereas *intention strength* refers to the degree of firmness a person expresses toward an intended action, *intention self-concordance* denotes the extent to which a specific goal intention is congruent with the basic needs, interests and values of the person” [21] (pp. 2–3, emphases in original). Their study already showed that physical activity can be explained by strength of intention as well as by its self-concordance [21]. However, it is an open question as to whether TPB components can be used to explain not only intention strength but also its self-concordance. Therefore, to better explain physical activity behavior, this paper proposes and tests a model that extends TPB by integrating self-concordance. Moreover, we investigate the potential moderating role of habit strength, assuming that TPB is more suited to explain nonhabitual than habitual behavior [22]. Thus, the paper contributes to the existing research both theoretically (by testing an enhanced model) and practically (by identifying salient behavioral determinants to inform a health promotion strategy in an increasingly important target group).

### 1.1. Theory of Planned Behavior and Physical Activity

The key assumption of TPB is that a person’s behavior (e.g., physical activity) is determined and best predicted by intention [23,24]. Accordingly, the stronger a person’s intention, the more likely a specific behavior will be performed. In turn, intention is determined by three components: a person’s attitude toward the behavior, the subjective norm, and PBC—the latter may also have a direct influence on behavior. Attitude is best described as the evaluation of a specific behavior and its perceived consequences. Subjective norm describes the perceived social pressure from important others to perform a behavior. Finally, PBC addresses the control beliefs, which include barriers and facilitators of performing a behavior. Other variables such as demographic attributes and general values are assumed to indirectly influence intention and behavior through a person’s behavioral beliefs [23].

Various meta-analyses confirmed that TPB can explain physical activity [25,26,27]. A more recent review of 11 studies concluded that TPB is also suited to explaining the physical activity of individuals with disabilities [28]. For older adults, it was found that those who held a positive attitude toward exercising had a high PBC and that those who perceived pressure from important others had a higher intention to exercise [29]. However, recent results on TPB in the context of older adults’ physical activity are mixed. While PBC and attitude are found to influence the intention to perform physical activity and the behavior itself, no effect was found for the influence of the subjective norm [18,19]. Therefore, our hypotheses concerning TPB components and intention strength are a positive relation for attitude (H1a) and PBC (H2a), and no relation for subjective norm (H3a) (see Figure 1).

TPB conceptualizes how people consciously deliberate attitudes, subjective norms, and PBC when forming intentions and deciding to engage in specific behaviors. Thus, habitual behaviors resulting from the repeated performance of a behavior and performed automatically without deliberating cognitions are beyond the scope of the theory [30]. However, it can be argued that habitual behaviors, as well, are the result of a rational behavioral decision in advance, indicating that they are based on the same cognitions as nonhabitual behaviors [12]. Thus, the influence of the cognitive determinants on habitual behavior will not vanish completely but will be weaker. Accordingly, research shows that the relation between the intention to engage in physical activity and the actual performance of physical activity is considerably stronger when people are not used to being physically active than when people have a strong habit of being physically active [31,32,33]. As research in other contexts shows, the same holds true for the relation between intention and its antecedents (e.g., [34]). Therefore, we hypothesize that the positive relation between attitude and intention strength (H1b) as well as PBC and intention strength, respectively (H2b), is stronger for people with a weak habit than for those with a strong habit, while there is no difference for subjective norm (H3b), as we do not expect any relation between the subjective norm and intention strength (see Figure 1).

### 1.2. Self-Determination Theory and Physical Activity

Another reason for the inconsistent findings may be that TPB does not distinguish between different qualities of intentions. Behavior may be driven by the strength of an intention, as defined in TPB, but also by the type of motivation (intrinsic or extrinsic) behind it [21,35]. Self-determination theory (SDT) provides a theoretical framework for explaining this motivation and its relation to behavior. It conceptualizes a motivation continuum of different forms of extrinsic motivation and intrinsic motivation [36]. Each type of motivation includes a specific type of regulation and perceived locus of causality. These characteristics determine whether a behavior is non-self-determined, self-determined, or located between these two forms. On one end of the continuum, intrinsic motivation is characterized by an inherent pleasure to perform an activity and is highly autonomous. On the other end of the continuum, four types of extrinsically motivated behaviors can be distinguished, varying with respect to the extent to which their regulation is autonomous [36]. When behaviors are performed solely because of external pressures, like reward, punishment, or social influences, they are completely externally regulated. Introjected regulation describes behaviors that are performed due to an inner pressure and to avoid negative feelings such as guilt and shame [37]. A third type of extrinsic motivation is labeled identified regulation, which is a more autonomous and self-determined form of extrinsic motivation. Here, an individual perceives an action as personally important. Finally, integrated regulation is the most autonomous form of extrinsic motivation and is characterized by regulations that are consonant with one’s convictions. They differ from intrinsic motivation with regard to the question of whether they are performed to attain a certain outcome rather than for pure enjoyment [36].

SDT has been successfully used to investigate behavior in many fields, including physical activity [38,39,40,41,42,43]. Some studies have already shown its relevance to the physical activity of older adults. One study found that older women who dropped out of a physical activity program showed lower levels of self-determined motivations than did persistent woman [44]. In investigating the physical activity of older adults, another study also found significant differences in intrinsic and extrinsic motivations between exercisers and nonexercisers [45]. In general, self-determination is highly relevant for our specific target group. While current research underlines the importance of self-determination for older adults’ well-being [46,47], aging accompanies distinct challenges that threaten people’s self-determination [48]. Thus, integrating SDT with TPB may improve our understanding of older adults’ motives associated with starting and maintaining physical activity.

### 1.3. Integration of Self-Concordance (SDT) into TPB

Previously, attempts have been made to integrate TPB and SDT in order to gain a better understanding of various health behaviors. A meta-analysis of 36 studies that integrated TPB and SDT variables to explain health behavior showed that self-determined motivation predicts behavioral intention, partially mediated by the predictors of TPB: attitude, the subjective norm, and PBC [35]. A row of studies with children and university students showed that the intention to be physically active was better explained when aspects of SDT were integrated into TPB-based models [49,50,51,52].

The idea behind integrating both theories is that general motives of SDT serve as sources of information for the formation of specific belief-based constructs of TPB that are set on a situational level [49]. However, integrating SDT and TPB constructs creates the risk of overlaps or redundancies, especially when one takes behavioral beliefs into account [26]. According to TPB, attitude, subjective norm, and PBC are influenced by a corresponding set of beliefs, namely (a) behavioral consequences and their evaluation, (b) social expectations of important others toward the behavior and the motivation to comply with these expectations, and (c) perceived control factors and their power to facilitate or inhibit the performance of the behavior [12]. These beliefs are set on a more general level and partially overlap with general motivations of SDT. For example, having fun while being physically active may be an indicator of intrinsic motivation as well as a positive behavioral consequence.

Therefore, we suggest another possibility related to the integration of self-determined motivation into TPB by drawing on the concept of self-concordance. Self-concordance describes the extent to which the behavioral intention is self-determined. This concept extends SDT by “focusing on people’s broad personal goal statements, rather than focusing on domain-specific motivations and situational factors that can influence it” [53] (p. 152). According to theory, self-concordance describes the extent to which an individual’s goal reflects enduring interests and personal values [54]. Transferring the idea of goal self-concordance to intention self-concordance, people’s voluntary choices, beliefs, and intentions can be addressed [21]. Accordingly, self-concordant goal intentions are pursued because of an intrinsic drive, while non-self-concordant goal intentions are pursued because people feel that they must engage in a certain behavior [55]. In general, there are four types of intention self-concordance: external, introjected, identified, and intrinsic. Both external and introjected goal intentions are defined as non-self-concordant, as they are pursued either with a feeling of external pressure or in consideration of specific norms that do not correspond with the individual’s personal beliefs. Identified and intrinsic goal intentions, on the other hand, are associated with free choice and the belief in a relevant outcome as well as an inherent interest in, and challenge within, an activity [54]. These self-concordant goal intentions are associated with a greater probability of goal attainment [56]. Accordingly, two dimensions of intentions can be differentiated: (a) intention strength, which is usually measured in TPB studies, and b) intention quality, which refers to self-concordance [21]. Within a short time frame of two to three weeks, one study with students did not find any influence of self-determined intention on their physical activity [20]. However, in a study with adults that investigated the prediction of physical activity through the interplay of intention strength and intention self-concordance, researchers found both intention strength and intention self-concordance to be significant and independent predictors of changes in physical activity over the period of one year [21]. However, it is still an open question as to whether TPB components can explain both aspects of behavioral intention. Accordingly, we integrate self-concordance of intention into our TPB-based model and hypothesize the following relations with attitude, PBC, and the subjective norm (see Figure 1).

We argue that a positive attitude and a high PBC imply self-determined regulations. Assuming positive behavioral consequences of being physically active and therefore holding a positive attitude will facilitate the experience of inherent pleasure in the activity and its perception as being personally important. Therefore, we hypothesize positive relations between attitude and intrinsic (H4a) and identified regulation, respectively (H7a). Similarly, the perception that there are no barriers to being physically active and the maintenance of a high PBC will increase the likelihood of being physically active for its inherent pleasure and personal importance. Hence, we assume positive relations between PBC and intrinsic (H5a) and identified regulation, respectively (H8a). Subjective norms, on the other hand, symbolize external pressure and may, therefore, inhibit self-determined regulations. Thus, we hypothesize negative relations between the subjective norm and intrinsic (H6a), as well as identified, regulation (H9a).

On the other hand, we assume that a negative attitude toward being physically active will result in non-self-determined intentions. If an individual holds a negative attitude, his or her intention to be physically active may rather be motivated by inner pressure or external sanctions. Therefore, we hypothesize negative relations between attitude and introjected (H10a) and external regulation, respectively (H13a). As stated above, we assume a positive relation between PBC and intrinsic, as well as identified, regulations, because the perception of the existence of strong barriers may undermine the notion of being physically active for its inherent pleasure. However, we do not assume any relationship between PBC and an inner pressure or the anticipation of external sanctions. Hence, we hypothesize no relation between PBC and introjected (H11a) and external regulation, respectively (H14a). As subjective norms symbolize this extrinsic pressure, we assume positive relations between the subjective norm and introjected (H12a), as well as external, regulation (H15a) (see Figure 1).

As TPB components may be better suited to explaining behavior with a weak habit (see above), we hypothesize that the expected relations are stronger for people with a weak habit than for people with a strong habit (H1b, H2b, H4b to H10b, H12b, H13b, and H15b) and that there is no difference for the nonexpected relation between PBC and introjected/external regulation (H3b, H11b, and H14b).

## 2. Methods

### 2.1. Design and Procedure

To test our proposed research model (see Figure 1), we conducted a standardized computer-assisted telephone interview (CATI) in October 2018. This study followed the Declaration of Helsinki [57] and was approved by the Advisory Board on Ethical Issues of the University of Erfurt. The basic population of this survey consisted of people 65 years and older living in Germany. Participants were selected by random digit dialing. If more than one person aged 65 years or older lived in a household, participants were chosen by the next-birthday method. Participation was voluntary and not incentivized. Participants gave consent prior to anonymously answering the survey. Data collection process was certified according to ISO 20252. Only anonymized data was stored and analyzed. The average interview length was 30 min. The minimum sample size for 80% test power on a significant level of 5% with a minimum *R^2^* of 0.10 and three independent variables would have been 103 [58] (p. 22). In total, 1001 participants finished the survey. Participants who did not answer a minimum number of the model-relevant questions (*n* = 40) were excluded. In addition, those having a very low behavioral intention to be physically active (below 3 on a scale from 1 = “no, not at all” to 5 = “yes, of course”) could not be considered in the analysis, because, due to a filter in the questionnaire, they did not answer the self-concordance questions (*n* = 96). Thus, the final sample size was *N* = 865.

### 2.2. Measures

TPB constructs were measured according to the recommendations of Ajzen [59], while self-concordance of behavioral intention was assessed using a scale developed by Seelig and Fuchs [60] in the context of physical activity. For most questions, answers were captured on a five-point Likert scale from 1 = “no, in no case” to 5 = “yes, in any case”. Participants were given the following definition of physical activity: “Generally speaking, all activities that make the heart beat faster and require more air, resulting in faster breathing, e.g., classic sports, but also gymnastics, walking, climbing stairs, vigorous housework, gardening, etc.” For intention strength, we asked, “Do you intend to be physically active several days a week in the near future?” with “being physically active several days a week” taking into account that many older adults suffer from physical limitations that make it difficult to exercise. Participants who indicated at least 3, “partly”, also responded to the self-concordance scale. This scale included three items for each of the four regulations that were introduced with “In the near future, I intend to be physically active several days a week because …”. Examples are “… I simply enjoy it” (intrinsic), “… I have good reasons” (identified), “… otherwise I would have to blame myself” (introjected), and “… other people tell me to be physically active” (external).

Perceived behavioral control was measured with one question about capacity (“Do you think you will be able to be physically active several days a week in the near future?”) and one question about autonomy (“It is in my power to be physically active several days a week in the near future”). To avoid sequence effects, other questions about the subjective norm and attitude were placed between these two. The measurement of the subjective norm included descriptive aspects (“People who are important to me are physically active several days a week”) and injunctive aspects (“People who are important to me think I should be physically active several days a week in the near future”). Attitude was operationalized with the affirmation of five attributes including both instrumental and experiential aspects of the behavior and the overall evaluation. In addition, three items on habit strength [61,62] were included—for example, “I feel weird if I don’t get to exercise”.

### 2.3. Data Analysis

To test the hypotheses of our conceptual research model simultaneously, we used structural equation modeling (SEM). Particularly, we chose Wold’s partial least squares (PLS) approach [63,64], because it is less stringent about the distribution assumption as compared to the alternative linear structural relationships approach (LISREL [65]). Moreover, PLS is recommended for novel theoretical models that have not been tested thus far [66]. The gradual statistical analysis was conducted with SmartPLS 3 (SmartPLS GmbH, Bönningstedt, Germany).

Initially, we evaluated the internal consistency of the reflective measurement models by reviewing the indicator reliability (factor loading ≥0.70; [67]) and the composite reliability (≥0.60; [58,68]). Within this process, two indicator variables of the constructs of attitude and external regulation had to be removed because of insufficient factor loadings and internal reliability. All other constructs showed satisfactory results (see Appendix A in the Appendix A). To evaluate the convergent validity, we tested the average variance extracted (AVE). All constructs met the minimum value of AVE ≥ 0.50 [58,66]. Third, we ensured the discriminant validity of the reflective measurement models by controlling for the Fornell–Larcker criterion [66]. Additionally, we checked for multicollinearity among variables. VIF values were within the acceptable thresholds [58]. After validation of the quality criteria of the measurement models, we computed the actual structural model to determine its explanatory power (explained variance *R^2^*, effect size *f^2^*), explanatory contribution (path coefficient), and predictive relevance (Stone–Geisser’s *Q^2^*). For the statistical analysis of the group comparison, we followed Hair et al.’s [58] recommendation for the PLS approach and chose the permutation test in SmartPLS (SmartPLS GmbH, Bönningstedt, Germany). This test randomly exchanges observations between two groups and re-estimates the model for each permutation [58]. From the statistical significance of the differences between the group-specific path coefficients, we can deduce whether there are distinctions in the connections of our research model between older adults with a weak and strong habit.

## 3. Results

### 3.1. Descriptive Results

#### 3.1.1. Full Sample

Participants were between 65 and 96 years old (*M* = 74.53, *SD* = 6.06). Half of them were female (50.5%), one in ten was still employed (9.6%), and most had earned a good education (13.2% university degree; 39.3% general qualification for university entrance; 24.0% general certificate of secondary education; 18.2% certificate of secondary education). Every second respondent had a chronic condition (49.1%). Nonetheless, their self-evaluated health status was quite good (*M* = 3.24, *SD* = 0.85, 1 = “bad” to 5 = “excellent”), and most of them did not feel notably restricted in their physical activity because of their health status (*M* = 2.27, *SD* = 1.20, 1 = “not restricted at all” to 5 = “very restricted”). This was also reflected in the active lifestyle of our respondents: On average, they had nearly six “active” days per week (*M* = 5.89, *SD* = 1.80) and had a strong habit of being physically active (*M* = 4.36, *SD* = 0.79, *α* = 0.71, 1 = “strongly disagree” to 5 = “strongly agree”).

Further, participants had a strong intention to stay physically active (*M* = 4.62, *SD* = 0.68) and felt quite self-determined in their motivation (intrinsic regulation: *M* = 4.15, *SD* = 0.90, *α* = 0.69, identified regulation: *M* = 4.51, *SD* = 0.72, *α* = 0.71). By contrast, the internal pressure (introjected regulation: *M* = 2.90, *SD* = 1.21, *α* = 0.69) and external pressure (external regulation: *M* = 1.65, *SD* = 0.99, *α* = 0.64) to be physically active were perceived as being notably weaker. Additionally, respondents had a positive attitude toward physical activity (*M* = 4.66, *SD* = 0.50, *α* = 0.72) and a high PBC to be active (*M* = 4.64, *SD* = 0.61, *α* = 0.51). The subjective norm (*M* = 3.28, *SD* = 1.20, *α* = 0.42) was the least pronounced TPB component.

#### 3.1.2. Subsamples

To evaluate the influence of habit strength on physical activity, we distinguished between respondents with a relatively weak habit (*M* = 1.00 to 4.22, *n* = 284, 33.3%); respondents with a very strong habit, who answered all three questions on the habit scale with the highest score (*M* = 5.00, *n* = 375, 44.0%); and respondents with an average habit (*M* = 4.23 to 4.99, *n* = 194, 22.7%). To test our hypotheses regarding differences between those with a rather weak and strong habit, we compared the two extreme groups (see Section 3.3). Participants in the two groups are comparable regarding age (weak habit: *M* = 74.05 years, *SD* = 6.07; strong habit: *M* = 74.74, *SD* = 6.00) and the proportion of men and women (weak habit: 44.0% female; strong habit: 42.9% female). However, among respondents with a weak habit, a higher percentage pronounced to have chronic diseases (53.9%) than among those with a strong habit (44.3%). This is also reflected in a significant difference in the self-evaluated health status (*t*(651) = −7.49, *p* < 0.001): older adults with a strong habit of physical activity evaluated their health status better (*M* = 3.48, *SD* = 0.84) than those with a weak habit (*M* = 2.99, *SD* = 0.82). Regarding the constructs of the theoretical model, respondents with a strong habit scored significantly higher on the measurements of attitude, perceived behavioral control, intention strength, intrinsic and identified regulations, and lower on external regulations than respondents with a weak habit (see Appendix A in the Appendix A).

### 3.2. Main SEM Analysis

The results of the main SEM analysis can be seen in Figure 2 and, in more detail, in Appendix A in the Appendix A. Of all endogenous variables in our model, intrinsic regulation (fun and enjoyment of being physically active) could be explained best (*R^2^* = 0.430), though, also, one third of the variance of identified regulation (*R^2^* = 0.320) and intention strength (*R^2^* = 0.313) were predicted through attitude, PBC, and the subjective norm. However, for introjected and external regulation, the explained variances were considerably lower: 15.6% and 7.7%, respectively.

#### 3.2.1. Intention Strength

To test our hypotheses, we examined the relationships between the predictors and various dimensions of intention. As postulated in H1a and H2a, intention strength was positively influenced by attitude (H1a, path coefficient = 0.135 ***, *f^2^* = 0.023) and PBC (H2a, path coefficient = 0.496 ***, *f^2^* = 0.305). However, the subjective norm did not have a significant influence on intention strength, supporting H3a (path coefficient = −0.059, *f^2^* = 0.005).

#### 3.2.2. Intrinsic and Identified Regulation

A similar pattern is revealed for the two self-concordant regulation forms. Attitude plays, by far, the most important role in the predicting of both intrinsic regulation (H4a, path coefficient = 0.536 ***, *f^2^* = 0.432) and identified regulation (H7a, path coefficient = 0.390 ***, *f^2^* = 0.192). Furthermore, PBC positively affected intrinsic regulation (H5a, path coefficient = 0.226 ***, *f^2^* = 0.076) and identified regulation (H8a, path coefficient = 0.275 ***, *f^2^* = 0.095). However, the subjective norm did not have the expected negative impact on intrinsic regulation (H6a, path coefficient = 0.008, *f^2^* = 0.000) and identified regulation (H9a, path coefficient = 0.081 **, *f^2^* = 0.010). In sum, an individual with a positive attitude toward physical activity and a high PBC is very likely to feel self-determined and find the reason for his or her active behavior within himself or herself. Accordingly, these four hypotheses (H4a, H5a, H7a, H8a) were accepted. However, feeling pressured by other people to become physically active (the subjective norm) did not negatively affect the self-determined motivation forms. Thus, H6a and H9a had to be rejected.

#### 3.2.3. Introjected and External Regulation

While attitude did not affect introjected regulation (H10a, path coefficient = 0.022, *f^2^* = 0.000), it negatively influenced external regulation (H13a, path coefficient = −0.133 ***, *f^2^* = 0.018). Thus, people with a highly positive attitude toward being physically active will perceive external pressures to a lesser extent. Accordingly, H13a was supported, and H10a was rejected. We further assumed that PBC does not have a notable influence on both non-self-concordant regulation forms. This holds true in the case of introjected regulation (H11a, path coefficient = −0.023, *f^2^* = 0.000). However, PBC had an unexpected negative influence on external regulation (H14a, path coefficient = −0.080 *, *f^2^* = 0.007). Thus, feeling in control of one’s own behavior (PBC) reduces the perception of external pressures (external regulation) to be physically active. Again, one hypothesis was supported (H11a), whereas the other (H14a) had to be rejected.

Lastly, when one looks at the two non-self-concordant regulation forms, the importance of the subjective norm becomes visible. The subjective norm had the strongest influence on both introjected regulation (H12a, path coefficient = 0.278 ***, *f^2^* = 0.083) and external regulation (H15a, path coefficient = 0.361 ***, *f^2^* = 0.154). Accordingly, older adults who feel strong social expectations to be physically active also perceive intense internal and external pressures in their motivational regulation. Thus, H12a and H15a were supported.

#### 3.2.4. Preliminary Summary

Overall, 11 of 15 hypotheses could be accepted (H2a, H3a, H4a, H5a, H7a, H8a, H11a, H12a, H14a) or—due to relatively small path coefficients and effect sizes (see Appendix A in the Appendix A)—at least partly accepted (H1a, H13a). These were mainly the supported relationships between attitude and PBC and intention strength, as well as the two self-concordant regulations (intrinsic and identified). For explaining the non-self-concordant regulations (introjected and external), the subjective norm came into play and proved to be a strong and significant predictor. Nevertheless, four hypotheses had to be rejected, because they either did not explain more variance (H6a, H10a) or—in the case of H9a and H14a—supported an opposite connection. These include mainly the connections between the subjective norm and intrinsic and identified regulation, where we could not find significant influences.

### 3.3. Group Comparison

In the second part of our analysis, we examined the influence of habit strength on the relationships between the model components. Therefore, we split the participants into three groups and selected the two extreme groups (very strong vs. weak habit) for further analyses (see Section 3.1.2). The results of the group comparison can be seen in Table 1 and are visualized in Appendix A in the Appendix A.

All endogenous constructs could be explained notably better when considering only people with a weak habit to be physically active. This was especially the case for their intrinsic regulation (*R^2^_weak habit_* = 0.388 vs. *R^2^_strong habit_* = 0.101) and identified regulation (*R^2^_weak habit_* = 0.281 vs. *R^2^_strong habit_* = 0.063), as well as for the strength of their intention (*R^2^_weak habit_* = 0.269 vs. *R^2^_strong habit_* = 0.175). In addition, the explained variance in introjected regulation (*R^2^_weak habit_* = 0.067 vs. *R^2^_strong habit_* = 0.060) and external regulation (*R^2^_weak habit_* = 0.179 vs. *R^2^_strong habit_* = 0.114) was larger for less active people than it was for agile ones. However, this disparity was less pronounced for the introjected and external regulation than for the first three endogenous variables. This large difference in explained variances revealed that our research model worked better for those who did not have a strong habit of being physically active.

#### 3.3.1. Habit Strength and the Relation between Attitude and Intrinsic/Identified Regulation

A closer look at the differences between the path coefficients in the two groups reveals details with regard to the various predictor variables. We found the strongest difference between the two groups in the influence of attitude on intrinsic regulation (H4b). The path coefficient was notably higher for people with a weak habit (path coefficient = 0.526) than for those with strong habitual behavior (path coefficient = 0.273). We found a similar, though less pronounced, pattern between attitude and identified regulation (H7b, path coefficient *_weak habit_* = 0.334 vs. path coefficient *_strong habit_* = 0.137). This difference was significant on the 5% level. Based on these results, we can assume that self-concordant motivations to become physically active can be better predicted by a positive attitude among people with a weak habit than for those with a strong habit. For these very active individuals who have already implemented the target behavior into their daily routines, the impact of a positive attitude on a self-determined intention is limited.

#### 3.3.2. Habit Strength and the Relation between Subjective Norm and Intrinsic Regulation

In the main SEM analysis, we could not find evidence that the subjective norm had the expected negative impact on intrinsic regulation (H6a). We further assumed that this negative effect would be stronger for people with a weak habit (H6b). Interestingly, though we had to reject our first assumption, we found a significant difference between both groups. While, for people with a strong habit, the subjective norm had a small but positive impact on the intrinsic motivation to be physically active (path coefficient = 0.093), this effect took a negative direction in the weak habit group (path coefficient = −0.048). As very active individuals will be surrounded by equally active people, the subjective norm—instead of exerting pressure to become more active—might have a positive impact on their self-determination. On the other hand, the perceived expectations of others might influence the self-determination of less active elderly. This statistical difference between both groups became visible only through the permutation test. Otherwise, it would have been overlooked.

#### 3.3.3. Other Findings/Tendencies

Lastly, we found two close-to-significant differences between both groups that can be regarded solely as tendencies (H2b, H15b). The first shows that the influence of the subjective norm on external regulation is greater for people with a weak habit (path coefficient = 0.414) than for those with a strong habit (path coefficient = 0.307). As assumed previously, these less active individuals might feel more pressure from their social peers, which leads to a more controlled, externally regulated motivation to enhance their activity behavior as compared to their agile counterparts.

The second tendency is revealed when one looks at the impact of PBC on intention strength, which is, again, larger for those with a weak habit (path coefficient = 0.513) than for those with a strong habit (path coefficient = 0.393). This indicates that people with a less active lifestyle can be more influenced to become physically active by improving their PBC as compared to people who are already active.

#### 3.3.4. Preliminary Summary

The analysis of group differences revealed five significant disparities between both groups (H2b, H4b, H6b, H7b, H15b). First, we found that the overall model worked best for less active people with a weak habit. In particular, self-concordant motivations to become physically active could be better predicted by a positive attitude among less active people than among agile ones (H4b, H7b). In addition, being in control of one’s own movement behavior (PBC) was an important predictor of the intention to become active in the future, in particular for people with a weak habit (H2b). Furthermore, while people with strong habitual behavior felt a slightly positive influence of the subjective norm on the enjoyment of physical activity, people with a weak habit perceived the subjective as being more of a hindrance to intrinsic motivation (H6b). Congruently, the influence of the subjective norm on external regulation was greater for less active people than it was for agile ones (H15b).

Lastly, the three hypotheses assuming no statistical differences in the path coefficients of both groups, regarding the impact of the subjective norm on intention strength (H3b), PBC on introjected regulation (H11b), and PBC on external regulation (H14b), could be supported. However, the seven remaining hypotheses had to be rejected, because there were no significant differences.

## 4. Discussion

Previous research on physical activity has already shown that it can be fruitful to combine TPB and SDT to fully understand the antecedents of this behavior [35,49,50,51,52]. Yet, this combination bears the risk of redundancies, as motivations and underlying beliefs of TPB constructs may overlap [26]. Accordingly, we chose another possibility to combine both theories. As described in chapter 1.3., Fuchs et al. [21] stated that two aspects of behavioral intention can be distinguished: strength and self-concordance, both of which showed a significant influence on physical activity. Accordingly, to promote physical activity in older adults, we must target a strong *and* self-concordant intention. However, they did not investigate TBP components as potential antecedents of both intention strength and its self-concordance. Therefore, we examined an innovative combination of TPB and SDT by integrating self-concordance of intention into a TPB-based model. This model was well-suited to explain the strength and self-concordance of older adults’ intention to be physically active. The influence of TPB components on intention would have been underestimated if we had investigated only its strength. Our results show that TPB components can and should be used to explain not only intention strength but also its self-concordance—not just because both predict actual behavior (as was shown in [21]) but also because intention strength and self-concordance differ in their relations with attitude, PBC, and the subjective norm.

In detail, the results of our study showed that the influence of attitude on intention would have been underestimated if we had focused only on intention strength. While the relation between attitude and intention strength was rather weak, there was a strong relationship with self-concordant intention, specifically self-determined regulations (intrinsic and identified) and especially for people with a weak habit. PBC had the strongest influence on intention strength. In addition, its positive relation with self-determined regulations and negative relation with external regulation underlined the high relevance of PBC for the intention to be physically active. The subjective norm was not related to intention strength—a result in line with those studies using TPB to explain the physical activity of older adults we already described in chapter 1.1. [18,19]. Thus, considering only intention strength would mask the role of the subjective norm as a predictor of physical activity. Taking self-concordance into account showed that the subjective norm had a strong positive relation with non-self-determined regulations (external and introjected). This means that high perceptions of subjective norms may undermine the self-determination of the intention to be physically active and, therefore, reduce the probability of the actual behavior.

Moreover, we took into account the habitualization of physical activity by comparing individuals with a weak vs. strong habit and, thus, found our model to be especially well-suited for people with a weak habit. This is an important finding both theoretically and practically. First, it is in line with the assumption that TPB is better suited to explain nonhabitual than habitual behavior [22,30]. Second, people with a weak habit are a relevant target group with an especially high potential to increase their physical activity. In this study, those with a weak habit also showed a larger proportion of chronic diseases and a lower self-evaluated health status. As multimorbidity increases with age [2], older adults with health constraints are an important target group and face specific challenges with regard to physical activity [69]. In addition, older adults are a very heterogeneous population [70]. Hence, target group-specific interventions should consider differences between subpopulations [71]. However, these differences may not only be based on sociodemographic characteristics but also on lifestyle segmentation [72,73].

## 5. Limitations

There are a few limitations. The first is the fact that our analysis was based on cross-sectional data. Hence, we were unable to predict actual (future) behavior. This means that we could not assess the relationship between behavioral intention and actual physical activity in older adults. However, Fuchs et al. [21] already showed that both intention strength and intention self-concordance significantly contribute to the prediction of future physical activity. Therefore, we can assume that the same holds true for our sample. Another limitation of our study is that our respondents, on average, had comparably active lifestyles. This can be explained by three aspects. Firstly, we assume a self-selection bias of respondents with a healthier lifestyle. Secondly, people with a very low intention to be physically active had to be excluded from the analyses, because they did not have to answer the self-concordance questions. Thirdly, although we selected respondents by random digit dialing, a greater number of well-educated and health-conscious people agreed to take part, which reduces the representativeness of our sample.

## 6. Conclusions

Our research yields important theoretical implications. We conclude that—when explaining intention to be physically active by attitude, PBC, and the subjective norm—not just intention strength but also its self-concordance should be examined. Combining SDT and TPB by integrating self-concordance into the level of behavioral intention in a TPB-based model provides a theoretical foundation that is better suited to explaining the intention to be physically active than does the use of only one of the theories. It would be interesting to investigate whether this holds true for other (health-related) behaviors, as well.

This study also provides important practical implications. Our results suggest that a high PBC and a positive attitude are key factors in the formation of a strong and self-determined intention. Accordingly, to promote physical activity among older adults, a high PBC and a positive attitude should be targeted. To foster a positive attitude, interventions should communicate those positive consequences older adults associate with being physically active. These may not only be physical health benefits but also psychological and social aspects of exercising such as increased psychological well-being and feeling related to other people [74]. To strengthen older adults’ PBC, their specific barriers and facilitators for being physically active should be addressed such as (lack of) exercise partners or social support [75,76]. Moreover, the potential negative impact of the subjective norm must be considered, as it may increase non-self-determined motivations that undermine the probability of being active in the long-term. Hence, messages promoting physical activity should support older adults’ perceived autonomy by providing choices, explaining the importance of the behavior and acknowledging conflicting personal feelings and perspectives [52].

## Figures and Tables

**Figure 1 ijerph-18-05759-f001:**
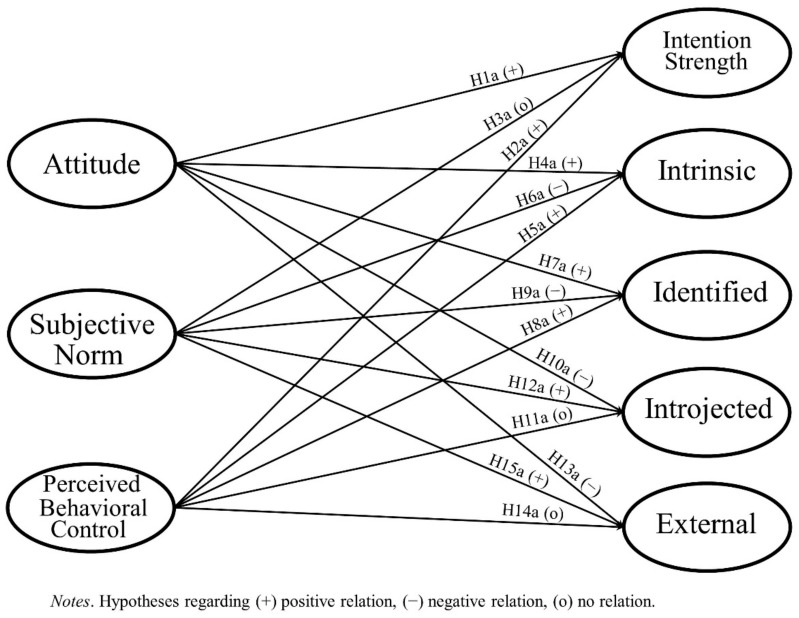
Proposed Research Model.

**Figure 2 ijerph-18-05759-f002:**
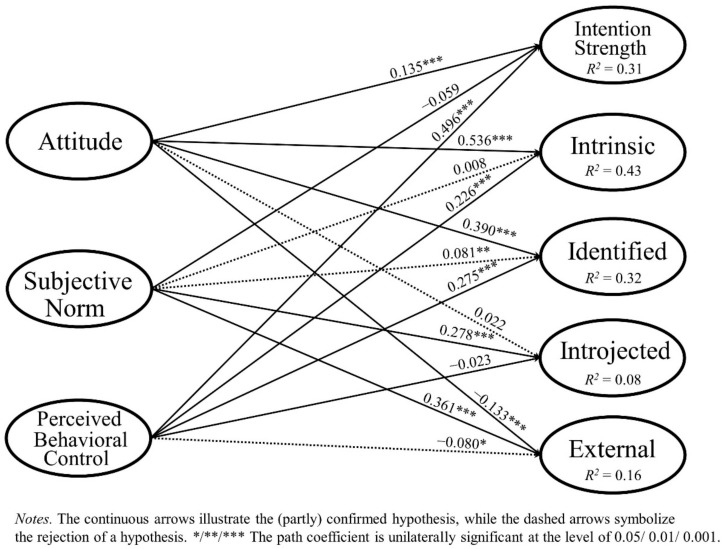
Results of the Main SEM Analysis.

**Table 1 ijerph-18-05759-t001:** Results of the group differences in the structural model.

Target Construct						
Impact Direction	Path Coefficient Group 1 (Weak Habit)	Path Coefficient Group 2 (Strong Habit)	Path Coefficient Difference	*R^2^* Group 1 (Weak Habit)	*R^2^* Group 2 (Strong Habit)	Hypothesis Testing
*Intention Strength*				0.269	0.175	
H1b: G1 (+) > G2	0.032	0.092	−0.060			x
H2b: G1 (+) > G2	0.513	0.393	0.120 ^(^*^)^			(√)
H3b: G1 (o) = G2	−0.050	−0.093	0.043			√
*Intrinsic Regulation*				0.388	0.101	
H4b: G1 (+) > G2	0.526	0.273	0.253 ***			√
H5b: G1 (+) > G2	0.227	0.103	0.123			x
H6b: G1 (−) > G2	−0.048	0.093	−0.141 *			√
*Identified Regulation*				0.281	0.063	
H7b: G1 (+) > G2	0.334	0.137	0.198 *			√
H8b: G1 (+) > G2	0.303	0.171	0.132			x
H9b: G1 (−) > G2	0.097	0.089	0.008			x
*Introjected Regulation*				0.067	0.060	
H10b: G1 (−) > G2	0.028	−0.053	0.080			x
H11b: G1 (o) = G2	−0.023	−0.016	−0.007			√
H12b: G1 (+) > G2	0.258	0.239	0.018			x
*External Regulation*				0.179	0.114	
H13b: G1 (−) > G2	−0.115	−0.113	−0.003			x
H14b: G1 (o) = G2	−0.084	−0.089	0.005			√
H15b: G1 (+) > G2	0.414	0.307	0.106 ^(^*^)^			(√)

*Note.* Explanation of hypotheses: G1 (+) > G2 positive relation that is stronger for G1; G1 (−) > G2 negative relation that is stronger for G1; G1 (o) = G2 no relation and no difference between groups. (*)/*/*** The path coefficient difference is unilaterally significant at the level of 0.10/0.05/0.001.

## Data Availability

The data presented in this study are openly available in OSF at https://osf.io/e856u/ (uploaded on 23 April 2021) with doi:10.17605/OSF.IO/E856U.

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
