# Peer review of "Determinants of Physical Activity in Older Adults: Integrating Self-Concordance into the Theory of Planned Behavior"

_ijerph, 2021, doi:10.3390/ijerph18115759_

Round 1

Reviewer 1 Report

The advantage of work is a novel approach that integrates protective physical elements in the intention to be physically active and a better habit to exercise. This can facilitate the paradigm shift from exercise-oriented goals to a personalized one that focuses more on the welfare of the elderly.

The older adult population is seriously described, but in the summary you could add important data such as the age range and the percentage of men or women.

In each of them, the psychometric properties of the measures used must be specified.

The results start directly with the SEM analysis. The comorbidities that can basically influence the results obtained are unknown. It would be advisable to add a small paragraph with these correlations.

The discussion is clear but should be a bit more supported by the literature proposed in the introduction.

In general, it seems to me a very interesting study with practical application, since it highlights the relevance of creating active physical exercise programs in older people, thus improving their physical and mental health. 

Reviewer 2 Report

This study investigated the key determinants of physical activity in older adults based on the theory of TPB and self-concordance. The combined model provides a better theoretical foundation from which to explain physical activity intentions than does just one of the theories. The results and conclusions are interesting and important for the areas of physical activity in older adults. Some minor suggestions may be helpful for the improvement of this study. 1. “2.3 . Sample characteristics and descriptive results.” The sample method and survey process should be described in the method section. The descriptive results about the participants may be better put into the results sections. 2. “4.4. Summary” paragraph should be put into the discussion section.

Reviewer 3 Report

Attachment
